# Guiding The Last Layer in Federated Learning with Pre-Trained Models

**Gwen Legate**
Concordia University, Mila
Montreal, Canada
`gwendolyne.legate@mila.quebec`

**Nicolas Bernier**
Concordia University, Mila
Montreal, Canada

**Lucas Caccia**
McGill University, Mila
Montreal, Canada

**Edouard Oyallon**
Sorbonne University, ISIR, CNRS
Paris, France

**Eugene Belilovsky**
Concordia University, Mila
Montreal, Canada

## Abstract

Federated Learning (FL) is an emerging paradigm that allows a model to be trained across a number of participants without sharing data. Recent works have begun to consider the effects of using pre-trained models as an initialization point for existing FL algorithms; however, these approaches ignore the vast body of efficient transfer learning literature from the centralized learning setting. Here we revisit the problem of FL from a pre-trained model considered in prior work and expand it to a set of computer vision transfer learning problems. We first observe that simply fitting a linear classification head can be efficient in many cases. We then show that in the FL setting, fitting a classifier using the Nearest Class Means (NCM) can be done exactly and orders of magnitude more efficiently than existing proposals, while obtaining strong performance. Finally, we demonstrate that using a two-stage approach of obtaining the classifier and then fine-tuning the model can yield rapid convergence and improved generalization in the federated setting. We demonstrate the potential our method has to reduce communication and compute costs while achieving better model performance. Code for our experiments is available[1].

## 1 Introduction

In recent years an increased focus on data privacy has attracted significant research interest in Federated Learning (FL). FL is an approach in which a common global model is trained by aggregating model updates computed across a set of distributed edge devices whose data is kept private. We desire to train a federated global model capable of equivalent performance as one trained on the same set of centralized data. However, it is worth noting that FedAvg [McMahan et al., 2017], the most commonly used FL baseline, has been shown to suffer from performance degradation when the distribution of data between clients is not iid [Li et al., 2020, Acar et al., 2021, Karimireddy et al., 2020].

Transfer learning from pre-trained models that have been trained on sufficiently abundant and diverse data is well known to produce state-of-the-art results in tasks related to vision [He et al., 2019, Girshick et al., 2014], Natural Language Processing (NLP) [Radford et al., 2019], and other domains. Indeed, pre-training combined with fine tuning to specialize the model for a specific downstream task often leads to better generalization and faster model convergence in the centralized setting [Weiss et al., 2016, Patel et al., 2015]. FL literature on the other hand, has been largely focused on models trained from scratch [McMahan et al., 2017, Karimireddy et al., 2020, Li et al., 2020] and the impact

---

[1]https://github.com/GwenLegate/GuidingLastLayerFLPretrain

of heterogeneity on algorithmic convergence. Recently, several studies have been conducted on the effect of pre-training on the performance of standard FL algorithms, see e.g., Chen et al. [2023], Nguyen et al. [2023]. Here, it was found that besides improving performance, pre-training can help to close the accuracy gap between a model trained in the federated setting and its non federated counterpart, particularly in the case of non-iid client data.

Prior work on transfer learning in the federated setting has focused on treating the pre-trained model as a stable initialization for classical FL algorithms that adapt all the parameters of the model. Approaches from the transfer learning literature demonstrate that it is often more efficient to adapt only parts of the model such as just the last layers [Kornblith et al., 2019], affine parameters [Lian et al., 2022, Yazdanpanah et al., 2022], or adapters Houlsby et al. [2019]. These approaches frequently yield combinations of better performance, computation time and better prevention against over-fitting. The selection is often a function of architecture, task similarity, and dataset size [Evci et al., 2022, Shysheya et al., 2023, Yazdanpanah et al., 2022]. In many supervised learning tasks studied in the literature such as transfer learning from ImageNet, the representations are powerful and training only the linear classifier is sufficient for strong performance [Kornblith et al., 2019].

A key driver of algorithmic development in FL is the reduction of communication cost, having largely motivated the design of the FedAvg algorithm (see e.g., McMahan et al. [2017]). Although not studied in the prior works, updating only the linear classifier can be highly efficient in the federated setting when starting from a pre-trained model. It can allow for both high performance, limited communication cost (since only the linear layer needs to be transmitted), and potentially rapid convergence due to the stability of training only the final layer. Training the linear classifier in a federated setting can lead to classical FL problems such as client drift if not treated appropriately. An example of this is illustrated in Nguyen et al. [2023, Appendix C].

We propose a two-stage approach based on first deriving a powerful classification head (HeadTuning stage) and subsequently performing a full fine-tuning of the model (Fine-Tune stage). Such two-stage approaches have been applied in practice and studied theoretically in the transfer learning literature [Kumar et al., 2022, Ren et al., 2023b]. They have been shown to give improved performance in both in-distribution and out-of-distribution settings [Kumar et al., 2022]. We highlight that the two-stage procedure can lead to many advantages in FL setting: (a) the fine-tuning stage is more stable when data is non-iid heterogeneous models, leading to substantial performance improvement (b) convergence of the fine-tuning stage is rapid (minimizing compute and communication cost)

For the HeadTuning stage, our work highlights the Nearest Class Mean (NCM), a classical alternative to initialize the classification layer which we denote FedNCM in the federated case. FedNCM can be computed exactly and efficiently in the federated setting without violating privacy constraints and we will demonstrate that in many cases of interest, using FedNCM to tune the classification head (without a subsequent fine-tuning) can even outperform approaches considered in prior work with significant communication and computation costs savings.

Our contributions in this work are:

- We provide empirical evidence that, for numerous downstream datasets, training only the classifier head proves to be an effective approach in FL settings.

- We propose employing a two-stage process consisting of HeadTuning (*e.g.*, via FedNCM or LP) followed by fine-tuning, results in faster convergence and higher accuracy without violating FL constraints. We further illustrate that it can address many key desiderata of FL: high accuracy, low communication, low computation, and robustness to high heterogeneity while being easier to tune in terms of hyperparameter selection.

- We present FedNCM, a straightforward FL HeadTuning method that significantly reduces communication costs when used, either as a stand alone technique, or as stage one (Head-Tuning) in our proposed two stage process which leads to improved accuracy.

## 2   Related work

**Federated Learning**   The most well known approach in FL is the FedAvg algorithm proposed by McMahan et al. [2017]. In the random initialization setting, convergence of FedAvg and related algorithms has been widely studied for both iid [Stich, 2019, Wang and Joshi, 2018] and non-iid settings [Karimireddy et al., 2020, Li et al., 2020, Fallah et al., 2020, Yu et al., 2019]. A commonly

cited problem in the literature is the challenge of heterogeneous or non-iid data and a variety of algorithms have been developed to tackle this [Li et al., 2020, Hsu et al., 2019, Legate et al., 2023, Karimireddy et al., 2020].

**Transfer Learning**    Transfer learning is widely used in many domains where data is scarce [Girshick et al., 2014, Alyafeai et al., 2020, Zhuang et al., 2020, Yazdanpanah et al., 2022]. A number of approaches for transfer learning have been proposed including the most commonly used full model fine-tuning and last layer tuning Kornblith et al. [2019] and some more efficient methods such as selecting features Evci et al. [2022], adding affine parameters Lian et al. [2022], Yazdanpanah et al. [2022], and adapters for transformers Houlsby et al. [2019]. Transfer learning and the effects of pre-training in FL have so far only been explored in limited capacity. In their recent publication, Nguyen et al. [2023] show that initializing a model with pre-trained weights consistently improves training accuracy and reduces the performance gap between homogeneous and heterogeneous client data distributions. Additionally, in the case where pre-trained data is not readily available, producing synthetic data and training the global model centrally on this has been shown to be beneficial to FL model performance [Chen et al., 2023].

**Nearest Class Means Classifier**    The use of the NCM algorithm in artificial intelligence has a long history. Each class is represented as a point in feature space defined by the mean feature vector of its training samples. New samples are classified by computing the distances between them and the class means and selecting the class whose mean is the nearest. In 1990, Ratcliff proposed to use NCM to mitigate catastrophic forgetting in continual learning and since then the use of NCM has been widely adopted and extended by continual learning researchers. This is due to its simplicity and minimal compute requirements to obtain a final classifier when a strong representation has already been learnt. Some of these methods include Rebuffi et al. [2017], Li and Hoiem [2017], Davari et al. [2022] who maintain a memory of exemplars used to compute an NCM classifier. Related to our work, recent literature in continual learning that have considered pre-trained models were shown to ignore a simple NCM baseline [Janson et al., 2022] which can outperform many of the more complicated methods proposed. In our work this NCM baseline, denoted as FedNCM for the federated setting, demonstrates similar strong performance for FL while serving as a very practical first stage of training in our proposed two-step process.

## 3    Methods

### 3.1    Background and Notation

In FL, distributed optimization occurs over $K$ clients with each client $k \in \{1, ..., K\}$ having data $\mathbf{X}_k, \mathbf{Y}_k$ that contains $n_k$ samples drawn from distribution $D_k$. We define the total number of samples across all clients as $n = \sum_{k=1}^{K} n_k$. The data $\mathbf{X}_k$ at each node may be drawn from different distributions and/or may be unbalanced with some clients possessing more training samples than others. The typical objective function for federated optimization is given in Eq. 1 [Konečný et al., 2016] and aims to find the minimizer of the loss over the sum of the client data:

$$\mathbf{w}^*, \mathbf{v}^* \in \arg\min_{\mathbf{w}, \mathbf{v}} \sum_{k=1}^{K} \frac{n_k}{n} \mathcal{L}(g(f(\mathbf{w}, \mathbf{X}_k), \mathbf{v})). \tag{1}$$

Here we have split the model prediction into $f$, a base parameterized by $\mathbf{w}$ that produces representations, and $g$, a task head parameterized by $\mathbf{v}$. In this work we will focus on the case where the task head is a linear model, and the loss function, $\mathcal{L}$ represents a standard classification or regression loss. The $\mathbf{w}$ are derived from a pre-trained model and they can be optimized or held fixed.

One approach to obtain the task head while using a fixed $\mathbf{w}$ is to optimize only $\mathbf{v}$ in a federated manner over all the data. In the case that $g$ is given as a linear model and we absorb the softmax into $\mathcal{L}$ this defaults to Linear Probing (LP)[Nguyen et al., 2023, Ren et al., 2023a].

### 3.2    FedNCM Algorithm

An alternative approach to derive an efficient $g$ is through the use of NCM. We note that FedNCM, the federated version of NCM, can be derived exactly in a federated setting. In FedNCM, outlined in Algo. 1., the server only communicates pre-trained weights once to each of the clients and the clients only communicate once with the server to send back their weighted class means. The server

---

**Algorithm 1 FedNCM**. $K$ is the total number of clients, $C$ is the number of classes in the training dataset, $D_c$ is the total number of samples of class $c$

---

**Require:** $(\mathbf{X}_1, \mathbf{Y}_1), (\mathbf{X}_2, \mathbf{Y}_2), \ldots, (\mathbf{X}_K, \mathbf{Y}_K)$ - Local datasets, $w_{pt}$ - pre-trained model

    **Server Executes:**
1: **for** each client $k \in K$ in parallel **do**
2:    $[m_c^k]_{c \in C} \leftarrow \text{LocalClientStats}(X_k, Y_k, \mathbf{w}_{pt})$    ▷ Send to all clients, receive weighted class means
3: **end for**
4: **for** each class $c \in C$ **do**
5:    $\mathbf{l}_c \leftarrow \frac{1}{D_c} \sum_{k=1}^{K} m_c^k$                       ▷ $\mathbf{l}_c$ can be used in NCM classifier
6: **end for**

    **Client Side:**
7: **function** LOCALCLIENTSTATS($\mathbf{X}, \mathbf{Y}, \mathbf{w}$)
8:    **for** each class $c \in N$ **do**
9:       Let $\mathbf{X}_c = \{x_i \in X, y_i = c\}$
10:      $m_c \leftarrow \sum_{x \in X_c} f_w(x)$
11:    **end for**
12:    **return** $[m_c]_{c \in C}$
13: **end function**

---

can then use each client's class means to compute the NCM exactly and use them either to perform classification directly using the class centroids, or to initialize a linear task head for further fine-tuning. FedNCM allows an efficient classifier approximation for pre-trained models and addresses many critical concerns in the FL setting including:

(a) **communication and computation**: FedNCM is negligible in both compute and communication, it requires one communication from the server to each client and back. Used as an initialization for further FT, FedNCM speeds up convergence and reduces the communication and computation burden

(b) **client statistical heterogeneity**: Robust to typical non-iid distribution shifts (not the case for LP or FT). A FedNCM initialization also makes further FT more robust to heterogeneity.

Notably, FedNCM can be computed using secure aggregation methods. Furthermore, the lack of update to the base model parameters naturally improves differential privacy guarantees [Cattan et al., 2022].

To use NCM as an initialization, consider the cross-entropy loss and $(g \circ f)(\mathbf{x}) = \mathbf{v} f(\mathbf{x}; \mathbf{w}) + \mathbf{b}$. We can set the matrix $v$ corresponding to the class $c$ logit with the normalized class centroid $\mathbf{l}_c / \|\mathbf{l}_c\|$ and the bias term to 0. This allows us to initialize the task head with FedNCM and obtain further improvement through fine-tuning $f$.

### 3.3 Two-stage Approach for Transfer Learning in FL (HeadTune + FineTune)

FL algorithms are often unstable due to the mismatch in client objectives which can lead to large changes during local training causing significant deviations amongst the different client models. When using a pre-trained model which allows us a powerful initial representation, we argue that a two-stage procedure will improve training stability and converge more quickly. In the first stage (HeadTune) we perform HeadTuning where the parameters of $g$ are updated *e.g.* by linear probing in federated fashion or by using FedNCM. As stated in Sec. 3.2, FedNCM is highly efficient, imposing a negligible cost in compute and communication with respect to any typical fine-tuning stage. In the second stage (FineTune), both $f$ and the classifier initialized in stage one, are fine tuned together in a federated setting according to the FL objective function specified in Eq. 1. Taking the negligible cost of communication and compute provided by FedNCM into account, our two-stage approach can have a substantial advantage in convergence when compared to simply a fine-tuning stage [Nguyen et al., 2023, Chen et al., 2023].

We now give an intuitive interpretation of the advantages of our method using the framework of Ren et al. [2023a]. Assume that the $k$-th worker is initialized via $\mathbf{w}_0$, and trained locally with SGD for several steps until it reaches the parameter $\mathbf{w}_k$. Writing $\mathbf{w}^*$ the optimal parameter, via triangular inequality, we obtain the following inequality:

$$\mathbb{E}_{\mathbf{X}_k}[\|f(\mathbf{w}_k; \mathbf{X}_k) - f(\mathbf{w}^*; \mathbf{X}_k)\|] \leq \mathbb{E}_{\mathbf{X}_k}[\|f(\mathbf{w}_0; \mathbf{X}_k) - f(\mathbf{w}^*; \mathbf{X}_k)\| + \|f(\mathbf{w}_k; \mathbf{X}_k) - f(\mathbf{w}_0; \mathbf{X}_k)\|]. \quad (2)$$

In the neural tangent kernel (NTK) [Ren et al., 2023a, Jacot et al., 2018] regime, for sufficiently small step size, Ren et al. [2023a] showed that the second term depends on the approximation quality of the head $g_0$ at initialization, which is bounded (where $\sigma$ is the sigmoid activation and $\{\mathbf{e}_i\}_i$ the canonical basis) for some $c > 0$, by:

$$\mathbb{E}_{\mathbf{X}_k}\|f(\mathbf{w}_k; \mathbf{X}_k) - f(\mathbf{w}_0; \mathbf{X}_k)\| \leq c \cdot \mathbb{E}_{(\mathbf{X}_k, \mathbf{Y}_k)}\|\mathbf{e}_{\mathbf{Y}_k} - g_\mathbf{v}(f(\mathbf{w}_0; \mathbf{X}_k))\|.$$

This suggests in particular that a good choice of linear head $\mathbf{v}$ will lead to a smaller right hand side term in Eq. 2, and thus reduce the distance to the optimum. Consequently, FedNCM or LP derived $\mathbf{v}$ (compared to a random $\mathbf{v}$) may be expected to lead to a more rapid convergence. Thanks to the initial consensus on the classifier, we may also expect less client drift to occur, at least in the first round of training, when $\mathbf{v}$ it initialized by HeadTuning, compared to a random initialization.

## 4   Experiments

In this section we will experimentally demonstrate the advantages of our proposed FedNCM and FedNCM+FT. Additionally, we show that simple LP tuning can at times be more stable and communication efficient than undertaking the full fine tuning considered almost exclusively in prior work on FL with pre-trained models.

Our primary experiments focus on standard image classification tasks. We also provide some NLP classification tasks in Sec. 4.2.1. We consider a setting similar to Nguyen et al. [2023] using the CIFAR 10 dataset [Krizhevsky, 2009] and expand our setting to include four additional standard computer vision datasets shown in Tab. 1. Following the method of Hsu et al. [2019], data is distributed between clients using a Dirichlet distribution parameterized by $\alpha = 0.1$ for our primary experiments. We set the number of clients to 100, train for 1 local epoch per round, and set client participation to 30% for CIFAR (as in Nguyen et al. [2023]). For all other datasets we use full client participation for simplicity.

| Dataset | Num. Classes | Num. Images |
|---|---|---|
| CIFAR-10 | 10 | 50000 |
| Flowers102 | 102 | 1020 |
| Stanford Cars | 196 | 8144 |
| CUB | 200 | 5994 |
| EuroSAT-Sub | 10 | 5000 |

Table 1: Summary of datasets used in our experiments.

Like Nguyen et al. [2023], we use SqueezeNet [Iandola et al., 2016], we also consider a ResNet18 [He et al., 2016] for experiments in Appendix E. When performing fine-tuning and evaluation for all datasets, we resize images to $224 \times 224$, the training input size of ImageNet. We run all experiments for three seeds using the FLSim library described in Nguyen et al. [2023].

**Baseline methods** We compare our methods to the following approaches as per Nguyen et al. [2023]: (a) *Random*: the model is initialized at random with no use of pre-trained model or NCM initialization. This setting corresponds to the standard FL paradigm of McMahan et al. [2017]. (b) *LP*: Given a pre-trained model, we freeze the base and train only the linear head using standard FL optimizer for training. (c) *FT*: A pre-trained model is used to initialize the global model weights and then a standard FL optimization algorithm is applied. (d) *LP and FT Oracles*: These are equivalent baselines trained in the centralized setting that provide an upper bound to the expected performance.

All of the above baseline methods as well as our FedNCM and FedNCM+FT can be combined with any core FL optimization algorithm such as FedAvg and FedAdam [Reddi et al., 2020]. Our experiments, we focus on the high-performing FedAvg, FedProx and FedAdam which have been shown to do well in these settings in prior art [Nguyen et al., 2023].

**Hyperpameters** We follow the approach of Nguyen et al. [2023], Reddi et al. [2020] to select the learning rate for each method on the various datasets. For CIFAR-10 and SqueezeNet experiments we take the hyperparameters already derived in Nguyen et al. [2023]. Additional details of selected hyperparameters are provided in Appendix B.

**Communication and Computation Budget** We evaluate the communication and computation costs of each proposed method. Costs are considered both in total and given a fixed budget for either communication or computation. For the communication costs, we assume that each model parameter that needs to be transmitted is transmitted via a 32-bit floating point number. This assumption allows us to compute the total expected communication between clients and server. It is important to

| Dataset | Method | Accuracy (%) | Total Compute ($\times$F) | Total Comm. (GB) |
|---|---|---|---|---|
| CIFAR-10 | Random | $67.8 \pm 0.6$ | $4.5 \times 10^8$ | 1803.71 |
| | FT | $85.4 \pm 0.4$ | $3.0 \times 10^7$ | 120.25 |
| | FedNCM+FT | $\mathbf{87.2 \pm 0.2}$ | $3.0 \times 10^7$ | 120.25 |
| | LP | $82.5 \pm 0.2$ | $1.0 \times 10^6$ | 0.82 |
| | FedNCM | $64.8 \pm 0.1$ | $\mathbf{1}$ | $4.1 \times 10^{-3}$ |
| CIFAR-10 $\times 32$ | Random (Nguyen et al. [2023]) | 34.2 | $1.5 \times 10^8$ | 601.24 |
| | FT (Nguyen et al. [2023]) | 63.1 | $1.5 \times 10^8$ | 601.24 |
| | FedNCM+FT | $\mathbf{67.9 \pm 0.4}$ | $7.5 \times 10^7$ | 300.62 |
| | LP (Nguyen et al. [2023]) | 44.7 | $5.0 \times 10^7$ | 4.10 |
| | FedNCM | $40.02 \pm 0.04$ | $\mathbf{1}$ | $4.10 \times 10^{-3}$ |
| FLOWERS-102 | Random | $33.2 \pm 0.7$ | $9.2 \times 10^6$ | 1916.76 |
| | FT | $64.5 \pm 1.0$ | $7.7 \times 10^5$ | 159.73 |
| | FedNCM+FT | $\mathbf{74.9 \pm 0.2}$ | $7.7 \times 10^5$ | 159.73 |
| | LP | $74.1 \pm 1.2$ | $5.1 \times 10^5$ | 20.93 |
| | FedNCM | $71.8 \pm 0.03$ | $\mathbf{1}$ | $4.2 \times 10^{-2}$ |
| CUB | Random | $15.0 \pm 0.7$ | $5.4 \times 10^7$ | 2037.18 |
| | FT | $52.0 \pm 0.9$ | $1.8 \times 10^7$ | 679.06 |
| | FedNCM+FT | $\mathbf{55.0 \pm 0.3}$ | $1.8 \times 10^7$ | 679.06 |
| | LP | $50.0 \pm 0.3$ | $9.0 \times 10^6$ | 122.88 |
| | FedNCM | $37.9 \pm 0.2$ | $\mathbf{1}$ | $8.2 \times 10^{-2}$ |
| STANFORD CARS | Random | $5.6 \pm 0.8$ | $8.6 \times 10^7$ | 2370.97 |
| | FT | $48.7 \pm 2.0$ | $2.4 \times 10^7$ | 677.42 |
| | FedNCM+FT | $\mathbf{54.8 \pm 1.2}$ | $2.4 \times 10^7$ | 677.42 |
| | LP | $41.2 \pm 0.5$ | $2.0 \times 10^7$ | 200.70 |
| | FedNCM | $20.33 \pm 0.04$ | $\mathbf{1}$ | $8.0 \times 10^{-2}$ |
| EUROSAT-SUB | Random | $85.8 \pm 2.7$ | $5.3 \times 10^7$ | 2104.32 |
| | FT | $95.6 \pm 1.0$ | $1.5 \times 10^7$ | 601.24 |
| | FedNCM+FT | $\mathbf{96.0 \pm 0.5}$ | $1.5 \times 10^7$ | 601.24 |
| | LP | $92.6 \pm 0.4$ | $5.0 \times 10^6$ | 4.10 |
| | FedNCM | $81.8 \pm 0.6$ | $\mathbf{1}$ | $4..10 \times 10^{-3}$ |

Table 2: Accuracy, total computation and total communication costs of pure HeadTuning methods (below dashed lines) and their full training counterparts. CIFAR-10 $\times 32$ indicates CIFAR-10 without re-sizing samples to $224 \times 224$, we include this setting so we can compare directly with values for FT, Random and LP reported by Nguyen et al. [2023]. We observe pure HeadTuning approaches, FedNCM and LP can be powerful, especially under compute and communication constraints. The unit, F, used to measure communication is one forward pass of a single sample, details of the communication and computation calculations are provided in AppendixA.

emphasize that linear probing only requires that we send client updates for the classifier rather than the entire model as is the case in the other settings. Consequently, LP has much lower communication costs when compared to FT for any given number of rounds. Our proposed FedNCM is a one-round algorithm and therefore has even lower communication costs than any other algorithm considered.

For computation time we consider the total FLOPs executed on the clients. We assume for simplicity that the backward pass of a model is $2\times$ the forward pass. For example, in the case of LP (with data augmentation) each federated round leads to one forward communication on the base model, $f$, and one forward and one backward (equivalent to two forward passes) on the head, $g$. Similarly, for FedNCM the communication cost consists only one forward pass through the data.

## 4.1 Efficiency of Pure HeadTuning for FL

As discussed in Sec. 1 tuning the classifier head is at times as effective or more effective than updating the entire model in the context of transfer learning Evci et al. [2022]. In prior work, this situation was briefly considered as a limited case in Nguyen et al. [2023, Appendix C.2] for CIFAR-10 and suggested that tuning just the linear head (LP) might be a weak approach in the heterogeneous setting. Here we first revisit this claim and expand the scope of these experiments to highlight where LP can be beneficial in terms of performance, communication costs, and compute time. Subsequently, we show another approach for approximating a good classifier, FedNCM, which can be competitive with orders of magnitude less computation and communication cost. We will demonstrate how to get the best of both HeadTuning and fine-tuning in the FL setting.

In Nguyen et al. [2023] the CIFAR-10 fine-tuning is done by feeding the $32 \times 32$ input image directly into a pre-trained ImageNet model. Since the architectures are adapted to the $224 \times 224$ size and trained at this scale originally, such an approach can lead to a very large distribution shift and may be

sub-optimal for transfer learning. Thus we additionally compare to CIFAR-10 using the traditional approach of resizing the image to the source data [Kornblith et al., 2019, Evci et al., 2022].

Tab. 2 shows accuracy, compute, and communication cost results for Pure HeadTuning Methods (FedNCM and LP) as well as full tuning approaches including our FedNCM+FT. We note that in Tab. 2, CIFAR-10-32 $\times$ 32 refers to results published in Nguyen et al. [2023]. We first point out the difference image input size has on the results and conclusion. Overall accuracy is much higher (highest is 86% vs 63%) and the gap between FT and LP is substantially smaller when using the model's native input size, it shows an absolute improvement of only $4.6\%$ vs $18.4\%$. For both sizes of CIFAR-10 CUB, Stanford Cars and Eurosat. FedNCM without FT can substantially exceed random performance while maintaining a highly competitive compute and communication budget. For the Flowers102 dataset, FedNCM can already far exceed the difficult-to-train FT setting and furthermore, LP alone exceeds both FedNCM and FT. Our two-stage method of FedNCM+FT outperforms all other methods in terms of accuracy. In what follows we will show how FedNCM+FT also allows high efficiency given a specific, potentially limited compute and computational budget. When considering the results, we note that CIFAR-10 contains the same object categories as the original ImageNet dataset but the Flowers102 and CUB datasets, represent more realistic transfer learning tasks and under these conditions we observe the true effectiveness of HeadTuning methods such as FedNCM and LP.

## 4.2 FedNCM then FineTune

We now study in more detail the two-stage approach described in Sec. 3.3. Fig. 1 shows the comparison of our baselines and FedNCM+FT with FedAvg. We show both accuracies versus rounds as well as accuracy given a communication and computation cost budget. Firstly, we observe that when going beyond CIFAR-10, LP can converge rather quickly and sometimes to the same accuracy as FT. This result shows the importance of HeadTuning and supports the use of LP in federated learning scenarios. Secondly, we can see the clear advantage of FedNCM+FT. After the stage one FedNCM initialization, it is able to achieve a strong starting accuracy and in stage two, it converges with a better accuracy than FT given the same computation budget. We note that FedNCM+FT for Cars and EuroSAT-Sub sees an initial performance decrease during the first few rounds of fine-tuning before rebounding to reach the best performance. This phenomenon is related to the chosen learning rate and further study is required to determine if this drop behaviour can be eliminated, potentially by using an initial warm-up period at a lower learning rate.

FedNCM+FT converges rapidly, allowing it to be highly efficient under most communication budgets compared to other methods. The second column of Tab. 2 shows that as we impose increasingly severe limitations on communication budgets, the performance gap between FedNCM+FT and all other methods widens significantly in favour of FedNCM+FT. Indeed for all the datasets FedNCM+FT is always optimal early on. For three of the datasets (Flowers, CUB, Cars) it exceeds LP over any communication budget. For CIFAR-10 and Eurosat LP can overtake it after the early stage, however, FedNCM+FT remains competitive and ultimately reaches higher performance. Similar trends are observed for computation time. We note overall as compared to FT the performance improvement can be drastic when considering the trade-off of accuracy as a function of communication and compute available. We also remark that the variance of LP and FedNCM+FT is lower across runs than the FT and Random counterparts.

We note that the Random baseline, typically requires longer training than others to reach the convergence criteria, thus for the purpose of our visualization we do not show the fully converged random baseline, which always requires many more communication rounds than the other approaches; however, the full curves are included in Appendix F.

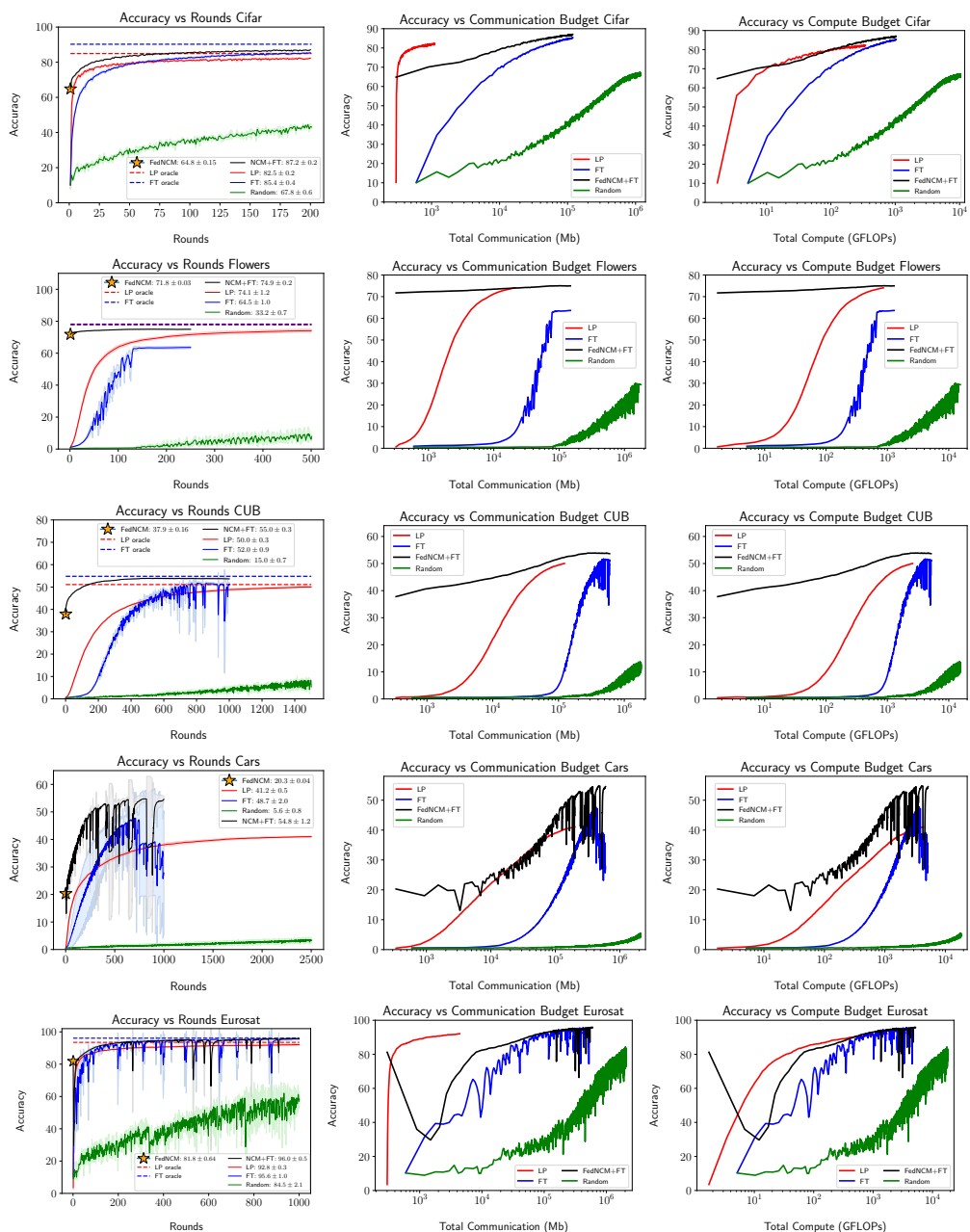

Figure 1: A comparison of the accuracy between models initialized with pre-trained weights and trained on different downstream tasks. The final result for both an LP and an FT oracle are shown and we remark that the NCM initialization allows the model to outperform the results of the LP oracle.

### 4.2.1 NLP Experiments

We now explore the effectiveness of our approach on NLP tasks. For these experiments, we train a DistillBert model [Sanh et al., 2019] on the AG News dataset [Zhang et al., 2015]. Fig. 2 shows the comparison of LP, FT and FedNCM+FT where we vary the number of examples per class (15, 90, 1470). We observe the strong performance of FedNCM as a stand alone method, reinforcing previous observations around the importance of HeadTuning. As models get larger, the importance of the final layer on the overall size of the model diminishes resulting in larger gaps in communication cost between one LP or FedNCM round and an FT round. Additionally, we find that FedNCM's accuracy is robust to sample size decreases and FedNCM+FT provides substantial convergence, and by extension communication cost benefits. Indeed it is well known that centroid based classifiers perform robustly in the case of few samples [Snell et al., 2017]. We provide additional confirmation of this in Appendix D with experiments using small subsets of Cifar-10.

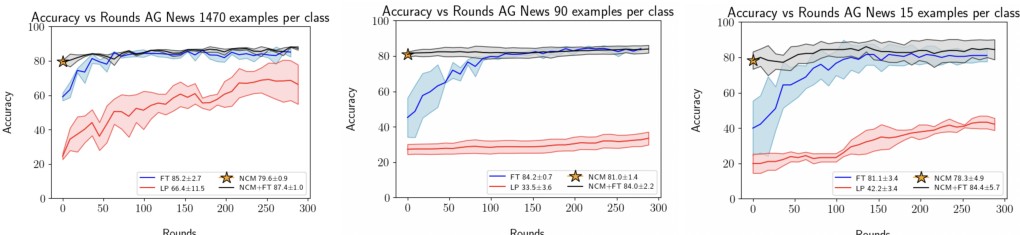

Figure 2: A comparison of the accuracy between models initialized with pre-trained trained using LP, FT and FedNCM+FT. We remark that the FedNCM alone obtains strong performance.

| Dataset | Algorithm | FedNCM | FedNCM + FT | FT |
|---------|-----------|--------|-------------|-----|
| CIFAR-10 | FEDAVG | $64.8 \pm 0.1$ | $87.2 \pm 0.2$ | $85.4 \pm 0.4$ |
| | FEDPROX | $64.8 \pm 0.1$ | $88.1 \pm 0.1$ | $87.8 \pm 0.08$ |
| | FEDADAM | $64.8 \pm 0.1$ | $\mathbf{89.4 \pm 1.1}$ | $88.2 \pm 0.2$ |
| FLOWERS102 | FEDAVG | $71.8 \pm 0.03$ | $74.9 \pm 0.2$ | $64.5 \pm 1.0$ |
| | FEDPROX | $71.8 \pm 0.03$ | $75.2 \pm 0.1$ | $65.5 \pm 1.5$ |
| | FEDADAM | $71.8 \pm 0.03$ | $\mathbf{76.7 \pm 0.2}$ | $66.6 \pm 1.0$ |

Table 3: Model performance with different methods for a variety of FL algorithms for FedAvg, FedADAM and FedProx. FedNCM+FT outperforms in all cases.

## 4.3 Analysis and Ablations

We now focus on demonstrating other advantages of our two-stage method, with a focus on using FedNCM as the method of choice for stage one (FedNCM+FT). In particular, we investigate robustness to larger number of clients, insensitivity to hyperparameters and compatibility with multiple FL algorithms and architectures. In Appendix C we ablate our two-stage method by comparing FedNCM, LP and a three-layer MPL as methods for stage one.

### 4.3.1 Choice of FL Algorithm

So far we have focused on comparisons using FedAvg as the baseline algorithm; however, since our method can be widely applied in FL, we further analyze FedNCM+FT using the FedAdam optimizer and the FedProx algorithm. Tab. 4.3.1 summarizes the results of using each method with 1.) FedNCM+FT and 2.) only FT, for the Cifar-10 and Flowers datasets.

We observe that improved FL optimizers can complement the two-stage FedNCM+FT which systematically exceeds the performance obtained when only fine-tuning. Regardless of the federated algorithm used, FedNCM alone continues to exceed the performance of FT on Flowers102. Our results suggest that choice of the FL optimization algorithm [Nguyen et al., 2023] is not always the most critical consideration for optimal performance when using pre-trained models.

**Hyperparameter Tuning** FL algorithms are known to be challenging for hyperparameter selection [Reddi et al., 2020] and this can affect their practical application. We first note that FedNCM does not have any hyperparameters which already provides a large advantage. In Fig. 5, we observe the final performance for a grid search over a range of server and client learning rates for FedAdam using both FT and FedNCM+FT. We observe that FedNCM+FT not only has higher performance but it is also more stable over the entire hyperparameter grid on Flowers dataset, and outperforms for all settings on CIFAR-10.

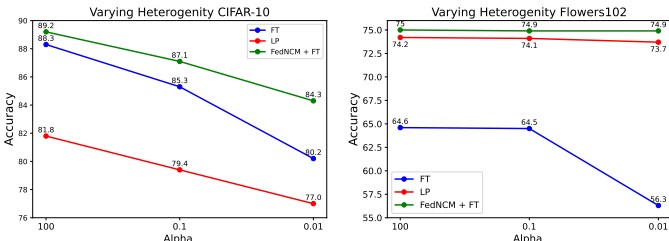

Figure 3: We vary the heterogeneity (Dirichlet-$\alpha$) for CIFAR-10 and Flowers102. Methods with HeadTuning: LP and FedNCM+FT are more robust, with the substantial advantage of FedNCM + FT increasing in challenging higher heterogeneity.

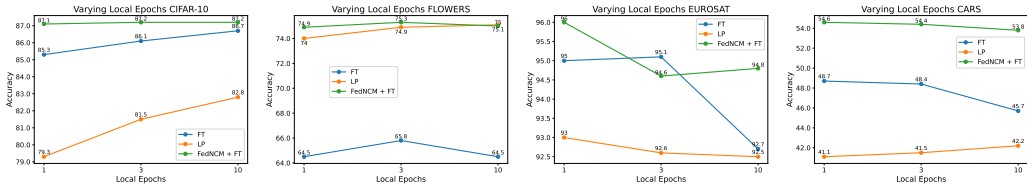

Figure 4: We vary the number of local epochs. FedNCM+FT always outperforms FT and nearly always LP in this challenging setting.

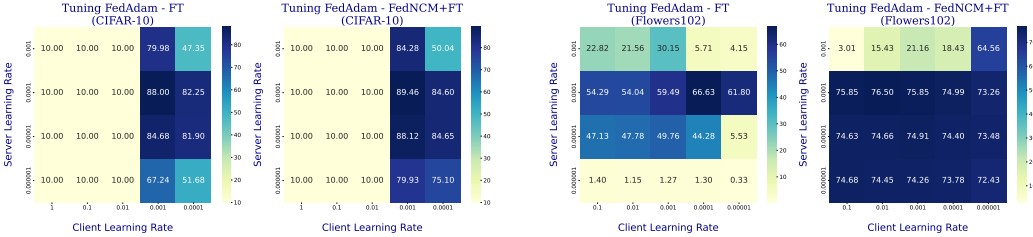

Figure 5: Hyperparameter grids for FedAdam for CIFAR-10 FT, FedNCMFT (left) and Flowers (right). We observe CIFAR-10 FedNCM-FT tends to do better or equal for all hyperparameters compared to FT. For Flowers it is much easier to tune, achieving strong values over a wide range, a noticeable advantage in FL

**Heterogeneity** Nguyen et al. [2023] points out that starting from a pre-trained model can reduce the effect of system heterogeneity. This is evaluated by comparing a specific Dirichlet distribution ($\alpha = 0.1$) used to partition data into a non-iid partitioning. Although the effect of heterogeneity is reduced we observe that in highly heterogeneous settings we still see substantial degradation in FT as shown in Fig. 3. Here we consider for CIFAR-10 the nearly iid $\alpha = 100$, $\alpha = 0.1$ as considered in Nguyen et al. [2023], and a very heterogeneous $\alpha = 0.01$. Firstly, we observe that FedNCM+FT can provide benefits in the iid setting. As heterogeneity degrades the naive FT setting sees a large absolute and relative drop in performance. On the other hand, FedNCM+FT as well as LP are able to degrade more gracefully.

**Varying the Local Epoch** The number of local epochs can drastically affect FL aglorithms, typically a larger amount of local computation between rounds is desired to minimize communication.

However, this can often come at a cost of degraded performance. We observe in Fig. 4 as in Nguyen et al. [2023] that FT can be relatively robust in some cases (CIFAR-10) to increasing local epochs. However, we also observe for some datasets that it can degrade, while LP and FedNCM+FT are less likely to degrade. Overall FedNCM+FT continues to outperform for larger local epochs.

**Increasing clients** Tab. 6 shows that as we increase the number of clients we observe that the degradation of FedNCM+FT is less severe than both LP and FT, suggesting it is stable under a large number of workers being averaged. As discussed in Sec. 3.3 it is expected in the same round that a representation would shift less from a starting point, and therefore since the starting point is the same for all clients, we expect the client drift within a round to be less given a fixed update budget.

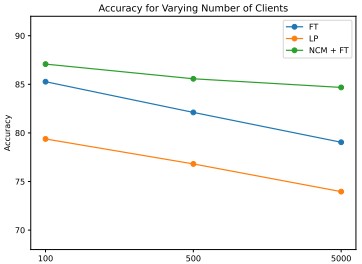

Figure 6: We increase the number of clients on CIFAR-10. FedNCM+FT degrades more gracefully than FT and LP.

## 5 Conclusion and Limitations

We have highlighted the importance of the last layers in FL from pre-trained models. We used this observation to then derive two highly efficient methods FedNCM and FedNCM+FT whose advantages in terms of performance, communication, computation, and robustness to heterogeneity were demonstrated. A limitation of our work is that it focus on image data and models, as this the primary set of data and models studied in prior work particularly in the context of transfer learning.

## Acknowledgements

This research was partially funded by NSERC Discovery Grant RGPIN- 2021-04104, FRQNT New Researcher Grant and CERC Autonomous AI. E.O. acknowledges support from ANR ADONIS - ANR-21-CE23-0030. We acknowledge resources provided by Compute Canada and Calcul Quebec.

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

| Dataset | FT+FedNCM | FT | LP | Random |
|---------|-----------|-----|-----|--------|
| CIFAR-10 | 200 | 200 | 200 | 3000 |
| CIFAR-10 32 × 32 Nguyen et al. [2023] | 1000 | 500 | 1000 | 1000 |
| Flowers-102 | 250 | 250 | 500 | 3000 |
| Stanford Cars | 1000 | 1000 | 2500 | 3500 |
| CUB | 1000 | 1000 | 1500 | 3000 |
| EuroSAT-Sub | 1000 | 1000 | 1000 | 3500 |

Table 4: rounds conducted for each dataset and method combination. For CIFAR-10 32 × 32 experiments were conducted in Nguyen et al. [2023] with the exception of FedNCM+FT (our method)

# Appendix

# A    Calculation of Communication and Computation Values in Tab. 2

## A.1    Communication

For HeadTuning methods such as LP and FedNCM, communication is calculated according to equation 3 where $LL_{in}$ and $LL_{out}$ are the input and output dimensions of the linear layer, $R$ is the number of federated rounds, $K$ is the total number of clients and $F$ is the fraction $F \in (0, 1]$ of clients participating in each federated round:

$$comm_{lp} = 2 * (LL_{out} * LL_{in}) * K * R * F * (32\ bit) \tag{3}$$

For methods training the complete model, communication is calculated according to equation 4 where $P$ is the number of parameters present in the base model (excluding the linear layer) and $LL_{in}$, $LL_{out}$, $R$, $K$ and $F$ have the same meanings as above:

$$comm_{ft} = 2((LL_{out} * LL_{in}) + P) * K * R * F * (32\ bit) \tag{4}$$

## A.2    Computation

For computation, let $F$ be one forward pass of a single sample, Let $S$ be the subset of clients, $K$, selected to participate in each federated round, $R$. We define one backward pass as $2F$ therefore each sample contributes $3F$ per round in a full fine-tuning situation. For HeadTuning a complete backward pass is not necessary since we only update the linear layer weights and each sample contributes $F$ per round. Eq. 5 and Eq. 6 formalize our computation calculation method in units of F, where $E$ is the number of local epochs performed by each client and $N_s$ is the sum of the number of samples at each selected client, $S$.

$$comp_{ft} = 3 * R * E * N_s \tag{5}$$

$$comp_{ft} = R * E * N_s \tag{6}$$

# B    Hyperparameter Settings

For CIFAR-10, CUB, Stanford Cars and Eurosat datasets the learning rates for the FedAvg algorithm were tuned via a grid search over learning rates $\{0.1, 0.07, 0.05, 0.03, 0.01, 0.007, 0.005, 0.003, 0.001\}$. For Flowers102, based on preliminary analysis we used lower learning rates were tuned over learning rates $\{0.01, 0.007, 0.005, 0.003, 0.001, 0.0007, 0.0005, 0.0003, 0.0001\}$. Tab. 4 summarizes the number of rounds conducted for each dataset and method combination.

Prior work on federated learning with pre-trained models has indicated that for FedADAM lower global learning rates and higher client learning rates were more effective. As a result for CIFAR-10 and Flowers the client learning rate was tuned over $\{1, 0.1, 0.01, 0.001, 0.0001\}$ and the server learning rate was tuned over $\{0.001, 0.0001, 0.00001, 0.000001\}$, each combination of server and client learning rates were tried.

| Dataset | HeadTuning Method | Accuracy |
|---|---|---|
| CIFAR-10 | LP | $82.5 \pm 0.2$ |
|  | MLP | $84.1 \pm 0.2$ |
| FLOWERS102 | LP | $74.1 \pm 1.2$ |
|  | MLP | $72.5 \pm 0.3$ |
| CUB | LP | $50.0 \pm 0.3$ |
|  | MLP | $50.3 \pm 0.3$ |
| CARS | LP | $41.2 \pm 0.5$ |
|  | MLP | $41.0 \pm 0.4$ |
| EUROSAT-SUB | LP | $92.6 \pm 0.4$ |
|  | MLP | $91.7 \pm 0.8$ |

Table 5: LP and MLP HeadTuning results prior to the fine-tuning stage.

| Dataset | stage 1 Method | Accuracy |
|---|---|---|
| CIFAR-10 | LP (5 ROUNDS) | $85.9 \pm 0.4$ |
|  | LP (10 ROUNDS) | $84.5 \pm 0.17$ |
|  | FEDNCM | $\mathbf{87.2 \pm 0.2}$ |
|  | N/A | $85.4 \pm 0.4$ |
| FLOWERS102 | LP (5 ROUNDS) | $68.6 \pm 1.3$ |
|  | LP (10 ROUNDS) | $68.3 \pm 0.6$ |
|  | FEDNCM | $\mathbf{74.9 \pm 0.2}$ |
|  | N/A | $64.5 \pm 0.1$ |

Table 6: Outcomes using LP as the HeadTuning method in our two stage training algorithm, FedNCM results from Tab. 2 are included for comparison.

# C    Additional HeadTuning Analysis

## C.1    MLP HeadTuning

We compare HeadTuning methods by replacing the linear layer in the LP method with a three layer MLP. Tab. 5 shows the HeadTuning results obtained using LP and MLP. We observe no advantage to using an MLP instead on a linear layer since most of our results are comparable or in some cases even inferior to the LP case.

## C.2    LP and FT

In this section we explore the use of LP as the HeadTuning method in our two-stage algorithm. Tab. 6 indicates performance of our two stage HeadTuning + fine-tuning method using five rounds of LP, ten rounds of LP and FedNCM as the HeadTuning methods. The bottom row in each dataset category where stage 1 Methods is denoted as n/a, is the result of only fine-tuning, *i.e.* not using our two stage method, which shows that HeadTuning prior to fine-tuning is always at least as effective as FT alone and, depending on the HeadTuning method selected, much more effective.

# D    Experiments Using Small Subsets of Data

We conduct experiments using only a small subset of Cifar-10 with nine samples of each class, *i.e.* 90 samples in total. We distribute these samples between five clients using a Dirichlet distribution with $\alpha = 0.1$ so each client will have very few samples and some will be entirely missing samples from many classes. Tab. 7 shows a $\geq 10\%$ improvement in accuracy between FT and FedNCM, which we attribute to overfitting of the FT model and the challenge of heterogeneity that FedNCM is not susceptible to. We also observe that LP does quite a bit better than FT indicating the benefit of HeadTuning methods in this setting.

# E    ResNet Experiments

We perform experiments for LP, FT, FedNCM and FedNCM+FT using the ResNet18 model using the FedAvg algorithm and the CIFAR-10 and Flowers102 datasets. Results are summarized in Tab. 8 where we observe that FedNCM performance is better by almost $13\%$ compared to squeezenet, while FT performance is degraded compared to squeezenet. We hypothesis this is due to the challenges of deeper networks in heterogenous federated learning. For the Flowers102 dataset, FedNCM and

| LP | FT | FedNCM | FedNCM + FT |
|---|---|---|---|
| $53.8 \pm 0.3$ | $45.2 \pm 2.4$ | $55.6 \pm 0.8$ | $56.7 \pm 0.7$ |

Table 7: ResNet18 model performance for FedAvg. As with Squeezenet, FedNCM+FT continues to outperforms in all cases.

| Dataset | FedNCM | FedNCM + FT | FT+Pretrain | LP+Pretrain |
|---|---|---|---|---|
| CIFAR-10 | $77.74 \pm 0.05$ | $79.05 \pm 1.31$ | $77.87 \pm 4.07$ | $74.73 \pm 3.03$ |
| FLOWERS102 | $74.13 \pm 0.31$ | $74.1 \pm 0.26$ | $34.41 \pm 10.16$ | $25.35 \pm 2.59$ |

Table 8: ResNet18 model performance for FedAvg. As with Squeezenet, FedNCM+FT continues to outperforms in all cases.

FedNCM+FT produce the best results by far. Additionally, for flowers FedNCM outperformed all other methods. The variance between runs using ResNet18 is much higher than was observed for SqueezeNet, FedNCM appears to help stabilize the results since it provides the most consistency by for both datasets.

# F   Extended Accuracy Comparison Figures

Fig. 7 is the extended version of Fig. 1 in the main body of the paper. In the paper we truncate the number of round displayed for the random setting since random requires many more rounds to converge than the other methods. Fig. 7 shows these same figures with the entirety of the training rounds displayed for the random setting.

# G   Compute

We use a combination of NVIDIA A100-SXM4-40GB, NVIDIA RTX A4500, Tesla V100-SXM2-32GB and Tesla P100-PCIE-12GB GPUs for a total of 1.1 GPU years . In addition to the experiments reported in the paper, this includes preliminary experiments and hyperparameter searches.

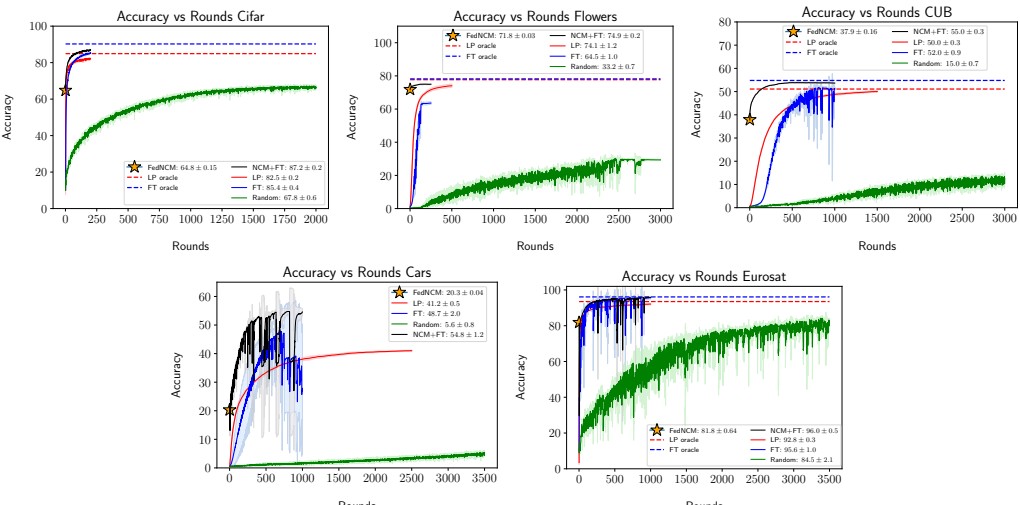

Figure 7: The full training of Random baseline corresponding to Fig. 1 in the paper is shown. We observer Random is always very far from the other baseliens and converges slowly.

