# OpenReview forum: "Guiding The Last Layer in Federated Learning with Pre-Trained Models"
_NeurIPS.cc/2023/Conference — NeurIPS 2023 poster_

### Official Review · Reviewer_1Br4 · 2023-06-30

**Soundness:** 3 good
**Presentation:** 3 good
**Contribution:** 3 good
**Rating:** 7
**Confidence:** 4

**Summary:**

The work proposes to address the training of the classifier (last layer of the network) separately in the setup of federated learning with a pretrained weights initialization. In particular, the suggestion is to use a nearest class means (NCM) approach, which saves a lot of computation time. The proposed approach is also supposed to be helpful in heterogeneous setup and thus evaluated on datasets separated into local ones by Dirichlet distribution. The paper provides an empirical evaluation of the linear probing and NCM approaches for training classifier, compared to standard federated learning. The conclusion made from the empirical results is that proposed method saves communication and leads to better performance.

**Strengths:**

The proposal to use only the last layer adaptation when the weights are pretrained is a viable and interesting idea for federated setup. It seems to mitigate heterogeneity and save computational time (with NCM) and communication (when no further finetuning is performed).

**Weaknesses:**

1) The explanation of the approach can be improved---especially the mixed one (is the final tuning happening for all the weights or without head?).

2) Heterogeneity is presented as one of the main reasons to introduce a new approach, but the method is not compared to any specific federated algorithms devised for it.

3) The evaluation is trying to highlight both LP and NCM, while overall they do have rather different approaches; this makes it unclear what exactly is the main message of the paper: to train only the last layer or to train it in the particular way? Moreover, only the combination with further tuning results in best performance.

Minor
- table3 is never referenced
- figure4 is referenced before figure3
- figure5 is called table5

**Questions:**

1) It seems that only head tuning + full tuning leads to the best results? But then it is not communication cheap?

2) It is claimed that NCM is more secure than standard FedAvg, but it is sharing the mean of the features for classes for each local learner. I think this can be a revealing information about local weights. Can you comment on this?

3) How the approach relates to layerwise personalization federated algorithms?

**Limitations:**

Evaluation for heterogeneity is done only with very particular non-iid with respect to labels, but it is not highlighted.

---

> ### Author Rebuttal · Authors · 2023-08-09
>
> We thank the reviewer for acknowledging benefits of our work and their helpful comments which help us improve the clarity of the paper. We address the comments in turn:
>
> **W1: is the final layer updated during fine-tuning**
> We apologize for the confusion, the final phase also updates the final layer head as in standard finetuning. We will clarify this in sec 3.3 on L156-157.
>
> **W2: Comparisons to methods addressing heterogeneity**
> Our work is based in the pre-trained FL setting, Ngyuen et al have already shown that in this setting, methods that have been developed particularly to address heterogeneity challenges in FL do not outperform FL algorithms not designed to address heterogeneity (*e.g.* FedAvg, FedAdam).  We thus focused on strong baselines for this setting that perform well, *i.e.* FedAvg and FedAdam. However, we have now done further experiments with FedProx, another high performing baseline from Nugyen et al that is designed to tackle heterogeneity. Please see results in the response to tHeD.
>
> **W3: Message**
> Our work has two related messages:
> 1) A  two stage method for training pre-trained models in a federated setting works best. The two stage method consists of first tuning the classification head (HeadTune) by any suitable HeadTuning method. In the second phase (FineTune) the entire model is fine tuned in a federated setting. In cases where compute or communication budgets are limited we highlight that using a HeadTuning method alone may be sufficient depending on performance requirements and the complexity of the downstream dataset
> 2) We propose a HeadTuning method, FedNCM that we demonstrate is highly efficient with regard to both communication, compute, and naturally robust to heterogeneity. We focus on the specific two-stage method where FedNCM is our chosen HeadTuning method which we follow by fine-tuning the entire model. We highlight this specific two-stage method since it achieves the best results overall.
>
> We agree the distinction could be clearer and will clarify this at the end of the introduction, we hope this will also help to address some of the ambiguity regarding the approach in your first point.
>
> **Q1: FedNCM+FT Not communication cheap**
> During full fine-tuning, all model weights are shared; however, in both NCM and LP only features, which represent a small  fraction of parameters, are shared by the clients. This means the cost of HeadTuning using NCM or LP is negligible compared to full fine-tuning. Despite this, in all cases HeadTuning causes the model to converge  faster as shown in Fig. 1, thus leading to a much better accuracy for a given communication budget.
>
> **Q2: NCM privacy/security**
> We argue that NCM and LP have less risk in terms of data privacy of the final FL trained model than FT trained models. This stems from the lack of updates to the base model parameters. Assuming the model is pre-trained on a public data, not updating the majority of the model leads to naturally improved DP guarantees (e.g. Tang et al, Cattan et al). We will clarify this in the text.
> Similar to gradient inversion attacks, FedNCM could be susceptible to an attempt by the attacker to invert the feature mean, however it is typically a challenging problem to invert even a single input sample from the last representation layer. For example, Dosovitskiy et al find it difficult to invert even for a shallow AlexNet layer. For FedNCM this inversion is even more challenging since unless a class has  only one sample at the client, a client the mean is transmitted which significantly dilutes individual class information and provides increased privacy.
>
> **Q3: personlized FL/layerwise**
> Our paper focuses on the non-personalized FL setting and starts from pre-trained models which is quite different to Ma et al work on personalized, layerwise FL. However, future work can consider personalization and combinations with layerwise update algorithms. Regarding the non-iid setting, we follow  a standard approach (and alpha value) from Nguyen et al, Reddi et al which selects a fixed alpha for the primary experiments. We also include results for other heterogeneity levels in the ablations.
>
> **References**
>
> [1] Tang, Qiaoyue, Lecuyer, Mathias. “DP-SGD-LF: Improving Utility under Differentially Private Learning via Layer Freezing”. 2023 https://openreview.net/forum?id=coLtCLTHFbW
>
> [2] Cattan, Yannis, et al. "Fine-tuning with differential privacy necessitates an additional hyperparameter search." arXiv preprint arXiv:2210.02156 (2022).
>
> [3] Dosovitskiy, Alexey, and Thomas Brox. "Inverting visual representations with convolutional networks." Proceedings of the IEEE conference on computer vision and pattern recognition. 2016.
>
> [4] Ma, Xiaosong, et al. "Layer-wised model aggregation for personalized federated learning." Proceedings of the IEEE/CVF conference on computer vision and pattern recognition. 2022.

---

> > ### Comment · Reviewer_1Br4 · 2023-08-11
> >
> > Thank you for the replies!
> >
> > Taking into account your W2 answer, can you please once again clarify the motivation for the new method for fighting heterogeneity: if in the setup with a pretrained model all the federated approaches are already good, why introduce another method? Is the main benefit then in saving communication in the initial phase and speeding up convergence?

---

> > > ### Author Response · Authors · 2023-08-13
> > > **Clarifications**
> > >
> > > Thank you for taking the time to read and reply to our rebuttal. First of all we want to clarify a potential misunderstanding: Nguyen et al  observed that in the pre-trained setting methods not designed for heterogeneity (e.g. FedAdam) can closely match and even outperform those that are designed for it, and also that the gap of iid vs non-iid is reduced (but importantly not closed). Despite performing well compared to other methods and outperforming random initialization, this does not mean that performance, communication efficiency, convergence speed, nor even robustness to heterogeneity is perfect, for example the performance in CIFAR-10 still lags behind centralized training.
> > >
> > > In our work our main message is about the importance of HeadTuning methods for achieving drastic improvements in multiple factors: communication, computation, convergence improvements, and often overall better performance. We note this can be a more critical factor than the choice of FL base algorithm.
> > >
> > > Robustness to heterogeneity however is another important observed side effect of our proposal. Firstly, FedNCM is completely robust to any of the typical non-iid data shifts (as opposed to LP or FT), and FedNCM+FT due to starting with a fixed head is more robust to heterogeneity than naive applications of FL algorithms. Note that **in our Figure 2 we consider more extreme non-iid case than in  Nguyen et al** with $\alpha$=0.01, here performance of base FL algorithm starts to degrade substantially, while with our two phase method the degradation is much more graceful

---

> > > > ### Comment · Reviewer_1Br4 · 2023-08-14
> > > >
> > > > Thank you for the reply.
> > > >
> > > > I am wondering if you somehow connect your research to model soups (Wortsman, Mitchell, et al. "Model soups: averaging weights of multiple fine-tuned models improves accuracy without increasing inference time." International Conference on Machine Learning. PMLR, 2022.)?
> > > >
> > > > Also, your comment on CIFAR10 reminded me of the known problem with averaging networks with batch normalization (e.g., Ainsworth, Samuel K., Jonathan Hayase, and Siddhartha Srinivasa. "Git re-basin: Merging models modulo permutation symmetries." arXiv preprint arXiv:2209.04836 (2022).) - maybe it is worth taking it into account.

---

> > > > > ### Author Response · Authors · 2023-08-15
> > > > > **Follow-up**
> > > > >
> > > > > Thank you for the reference to Model soup, we will add it. Notably Wortsman et. al. focus on fine-tuning on a *single* dataset, observing that combining models finetuned with different hyperparameters can be beneficial. This is distinct from our setting where we simultaneously fine-tune models with federated constraints on as many as 100 heterogeneous datasets. Their method also requires a centralized validation set. It can however be a potential future work to combine multiple models fine-tuned by our rapid fine-tuning procedure with different FL hyperparameters. However this would come at a higher communication and computational cost, although may be practical in some settings. We leave this exploration for future work.
> > > > >
> > > > > Regarding the connection between the issue we observed on L233 with respect to cifar-10 and normalization in git-rebasin. We want to emphasize the issue we observed is not unique to the federated learning or model averaging situation and can be observed in a standard centralized setting. Models trained on Imagenet are learning to extract features with input at 224x224 resolution, thus feeding heavily downsampled images for feature extraction can lead to distortion in the natural image hierarchy of CNNs, leading to large gaps in the linear model (or in NCM) performance when using the features.
> > > > >
> > > > > We note our image experiments do not use normalization layers as squeezenet the pre-trained model we focus on (inspired by Nguyen et al) does not have any. Our new NLP experiments with Distillbert use layernorm as this is part of the pre-trained model.
> > > > >
> > > > > We hope that we have addressed the concerns of the reviewer.

---

> > > > > > ### Comment · Reviewer_1Br4 · 2023-08-21
> > > > > >
> > > > > > Thank you for the discussion. I am therefore raising my score.

---

### Official Review · Reviewer_tHeD · 2023-07-05

**Soundness:** 4 excellent
**Presentation:** 4 excellent
**Contribution:** 3 good
**Rating:** 6
**Confidence:** 5

**Summary:**

This paper studies strategies to fine-tune models in Federated Learning, starting from a pre-trained model. It has been shown in the literature that pre-training is beneficial in FL, that it improves convergence speed and robustness to heterogeneity.

This paper shows that existent fine-tuning pipeline are not optimal. Further, this paper proposes a two stage head-tuning + fine-tuning technique, where the first head-tuning stage is compute and communication efficient. The authors show that their proposed two state approach are better than Linear Probing and full Fine-Tuning in terms of convergence to accuracy and  robustness to hyperparameters.

**Strengths:**

1. Experiments are done with rigor and described in detail the paper.
2. The authors ran into a discrepancy in terms of experiment results on one dataset reported in (Nguyen, 2023). They provided a concise reason and experiments to explain the source of discrepancy, which is due to the image size in the training data.


**Weaknesses:**

1. I would like to see how the proposed method compared to other simple two-stage fine-tuning methods, for example, running LP for a few epochs then perform full FT. This simple modification is not compared in the experiments.

2. high-dimensional classification: the output dimension on all problems studied in the experiments are not too big. In some high-dimensional classification problem, e.g. language model, some classes only have a few examples. I would like to understand how they affect the centroid-based initialization and the applicability of FedNCM.

**Questions:**

In Cars and Eurostat dataset, FedNCM + FT starts with the highest initialization accuracy, but the accuracy during training decreases first. Non-monotonicity are expected due to the stochastic nature of optimizer, but looking at Figure 1, the non-monotonicity here are more than what is produced by stochasticity.  In particularly Eurostat, the accuracy decreased for the first 1000 rounds to half of the starting accuracy before it improves.

I would like to understand more about this behavior. In practice when you are running FL in production where communications are costly, if one observes the accuracy continues to drop for the first 1000 rounds, the FL job will be killed and some bug will be suspected somewhere.

---

> ### Author Rebuttal · Authors · 2023-08-09
>
> We thank the reviewer for their overall positive view of the work and their helpful comments which will improve the paper. We address the comments in turn:
>
> **Weakness 1: Comparisons to first stage LP**
> We have added a comparison below with the suggested baseline of running LP for several rounds. We observe that FedNCM outperforms in the end while being simpler in the sense of not requiring additional communication rounds (and the associated overhead). It is an interesting idea to consider other Head Tuning approaches in our proposed two phase procedure in FL. We note the advantage of FedNCM, besides only taking one round, is that it naturally avoids heterogeneity and hyperparameter search issues which come with using LP or other HeadTuning approaches.  We will include these ablations in the appendix.
>
> |DS           |        5 rounds LP + FT   |         10 rounds LP + FT      |   FedNCM + FT     |
> |-----------|--------------------------------|------------------------------------|--------------------------|
> |Flowers |        68.6 (1.3)                  |       68.3 (0.6)                       |    74.9 (0.2)              |
> |Cifar-10 |        85.9(0.4)                   |       84.5 (1.7)                       |   87.2 (0.2)                |
>
> **Weakness 2: FedNCM with few samples / NLP Tasks**
>
> **Few Samples:**
> FedNCM has substantial benefits in the case of few samples as centroid based classifiers are often used in such settings and can perform robustly [1]. To confirm this we provide the results of an experiment using only a small subset of Cifar-10 with nine samples of each class, *i.e.* 90 samples in total. We distribute these samples between five clients using a Dirichlet distribution with $\alpha=0.1$ so each client will have very few samples and some will be entirely missing samples from many classes. We observe a $>10$% improvement in accuracy between FT and FedNCM, which we attribute to overfitting of the FT model and the challenge of heterogeneity that FedNCM is not susceptible to. We also observe that LP does quite a bit better than FT indicating the benefit of HeadTuning methods in this setting.
> |LP             |FT             | FedNCM      | FT+FedNCM |
> |--------------|--------------|  ----------------  | ----------------|
> |53.8 (0.3) | 45.2 (2.4) |    55.6 (0.8)  |    56.7 (0.7) |
>
> **NLP:**
> We also take the reviewer’s suggestion of considering an NLP problem and extend our results to consider several NLP tasks with a distillbert base. The results are shown in the accompanying pdf, and we find that FedNCM’s accuracy is robust as the sample size decreases and FedNCM+FT gives large benefits in convergence and thereby communication costs.
>
> [1] Snell, Jake, Kevin Swersky, and Richard Zemel. "Prototypical networks for few-shot learning." Advances in neural information processing systems 30 (2017).
>
> **Question: Why rapid initial drop in some cases**
> The behaviour you’ve mentioned in Cars and Eurosat where accuracy will decrease during the first few rounds of fine-tuning before rebounding and continuing to improve is related to the chosen learning rate from the hyperparameter search, which uses the standard setup for hyperparameter search (e.g. Nguyen et al and Reddit et al) that selects the best performing final accuracy on the validation set. Based on our results we are still able to achieve better performance than with any other method, but further study is required to determine if this drop  behaviour can be eliminated using an initial warm-up period at a lower learning rate. For a production system that may have other constraints than final accuracy, a different criteria can be used for the hyperparameter selection;

---

> > ### Comment · Reviewer_tHeD · 2023-08-21
> >
> > I appreciate the follow-up on the issues by the authors.
> >
> > Here are my comments regarding the author response:
> >
> > 1. response to weakness 1"Comparisons to first stage LP": this concern is sufficiently addressed.
> >
> > 2. response to few shot cases in NLP: the question is partially addressed. The authors provided experiments on AG-news classification tasks, and additionally considered setups with few examples per class. I want to clarify that, what my original suggestions was to consider cases where heterogeneity is high, and some classes has very few examples (extremely unbalance classification with some classes with only a few examples).
> >
> > Overall, my rating for the paper is unchanged.

---

### Official Review · Reviewer_8wWr · 2023-07-12

**Soundness:** 3 good
**Presentation:** 2 fair
**Contribution:** 2 fair
**Rating:** 5
**Confidence:** 3

**Summary:**

This paper proposes to use nearest class means (NCM) with pre-trained models in federated learning, coined as FedNCM. Experiments show that FedNCM is effective in terms of convergence and communication cost, due to the application of pre-trained models and light-weighted last layers.

**Strengths:**

1. Using pre-trained models in federated learning is a new and interesting direction, especially with the popularity of foundation models. This paper explores pre-trained models from a federated learning perspective, which shows the effectiveness of saving communication.
2. It provides code and experimental results on several vision datasets, including CIFAR-10, Flowers102, Stanford Cars, CUB and EuroSAT-Sub.
3. The experimental results show the improvement over Random baseline.

**Weaknesses:**

1. The algorithm seems to be a mere combination of NCM and FedAvg, which is not novel enough. This is not vital though as long as the performance is great.
2. However, the application of pre-trained models in FL has already been proposed in Nguyen et al. ICLR 2023.
3. The experimental comparison is not considering Federated Learning baselines, like FedProx, SCAFFOLD, FedYogi, LG-FedAvg [1], FedPer [2], etc.
4. The experiments only include small scale datasets, in total less than 50,000.
5. There is not much theory on the communication cost, or convergence rate analysis.
6. Minor: Line 201 FedAVG -> FedAvg; Line 234: imagenet -> ImageNet;


[1] Liang, Paul Pu, et al. "Think locally, act globally: Federated learning with local and global representations." arXiv preprint arXiv:2001.01523 (2020).
[2] Arivazhagan, Manoj Ghuhan, et al. "Federated learning with personalization layers." arXiv preprint arXiv:1912.00818 (2019).

**Questions:**

1. Line 165: Could you explain NTK or add a reference?
2. Could you explain more clearly the notations: FedNCM, FedNCM + FT, FedNCM + FT Pretrain, LP Pretrain, Random? Otherwise, the readers can only guess from these words.
3. Could you add more FL baselines as well as recent works on FL + pretrained model finetuning? For example, why is NCM preferable compared to softmax classifier?
4. Could you add more explanation about Fig 1, such as the different behavior on different datasets?

**Limitations:**

The limitation discussion is a bit limited and the authors just spend one sentence on this. Suggestions include the negative societal impact of using pretrained models and the lack of theoretical analysis and large-scale datasets, etc.

---

> ### Author Rebuttal · Authors · 2023-08-09
>
> We thank the reviewer for their helpful comments which help us to improve the paper, we address each comment in turn:
>
> **Clarification:** Our results show substantial improvement over prior work, not just the random baseline.
>
> **W1: Novelty**
> We emphasize that the simplicity of the proposed approach is one of its strengths. The judgment of novelty is subjective. We argue the observation that NCM is a powerful technique in the pre-trained FL setting and can be implemented under FL constraints. Moreover, the combination with a second stage of fine-tuning is not obvious and therefore quite novel.
>
> **W2: Prior work on pretrained models (Nguyen et al)**
> While it is true that Nguyen et al address the application of pre-trained models to FL, the body of work in this domain is still limited. Critically, our work highlights a very important and practically relevant limitation of the considerations of prior work. Specifically, both Nguyen et. al. and Chen et al  use pre-trained models as an initialization point and subsequently adapt **all** model parameters via fine-tuning. They observe for example that the choice of FL algorithm is less critical. Our work shows that there are more important considerations for pre-trained FL (*e.g.* only adapting the classifier, using a 2 phase approach) which can yield drastic improvements and supersede the choice of FL algorithm.   An additional benefit of our work is the use of a greater variety of datasets showing a greater diversity of results, thus exposing the limitations of only fine-tuning.
>
> **W3: Baseline FL algorithms**
> The baselines we chose are based on the prior work of Nguyen et al which shows (a) many FL methods achieve similar performance in the pre-trained setting (b) FedAdam and FedAvg were found to be strong performing baselines. Our preliminary experiments found that SCAFFOLD performs very poorly, thus it was excluded from subsequent analysis. Additionally, we note that FedPer is a personalization method while we focus on the standard shared model setting of FL.
>
> We agree that convincingly showing our method can work with a variety of base FL algorithms is beneficial, thus we now provide the results  for FT and FedNCM+FT using FedProx  (another well performing algorithm from Nguyen et al’) as the baseline method. We thank the reviewer for this suggestion which strengthens our evaluation.
> ## FedProx
> |     DS         |       FT        |       FedNCM+FT|
> |----------------|-----------------|-----------------------|
> |Cifar-10     | 87.8 (0.08)    |88.08 (0.1)|
> |Flowers     | 65.5 (1.5)      |75.2 (0.01)|
>
> FedProx performs similarly to FedAdam in Table 3 and our method substantially improves performance. We will include these results in our camera-ready.
>
> **W4: dataset scale**
> We use a variety of datasets consistent with previous work on pre-trained models. Our use of Cifar-10 allows us to make a direct comparison with Nguyen et. al., the best known work on federated transfer learning. We have also expanded our results to include some text datasets (see pdf), some with up to 120K samples.
>
> **W5: minimal theory on communication and convergence rate**
> We note that since we use a standard base FL algorithm, we will typically inherit its convergence analysis as a crude upper bound. Similar to the prior work in the pre-trained setting, our work focuses on practical empirical concerns. We leave further theoretical convergence rate analysis of fine-tuning and two stage training to future work. Also see more detailed response to 63Tk
>
> **W6: minor typos**
> Thank you for pointing out these inconsistencies, we will fix them in the final version.
>
> **Question 1:**
> NTK refers to the neural tangent kernel (Jacot et al, Ren et al), a recent tool for analyzing neural networks under assumptions on their training dynamics. In our work, we follow Ren et al and use NTK regime assumptions to show that initializing the linear head using FedNCM results in a smaller distance after fixed number of steps to the optimal solution than a random initialization.
>
> [1] Jacot, A., Gabriel, F., & Hongler, C. (2018). Neural tangent kernel: Convergence and generalization in neural networks. Advances in neural information processing systems, 31.
>
> **Question 2:**
> FedNCM is defined in section 3.2 (L134), it is the federated version of NCM and the algorithm is provided in Algo. 1. Random, LP and FT are defined in L191-L199 under the header Baseline Methods and the terms LP and FT Pretrain are simply LP and FT with “pretrain” added to explicitly indicate that we begin from pre-trained models. FedNCM+FT is our proposed two stage training where phase 1 →  FedNCM (HeadTuning) and phase 2 → FT (fine-tuning). In the final version we will make our explanation of the two stage method clearer and clarify or omit the use of “pretrain” in conjunction with FT and LP.
>
> **Question 3:**
> There has been limited work using pre-trained model initializations in FL, the best known reference is Nguyen et al with which we compare our work extensively. We have extend the base FL algorithms to include FedProx.
> **Why NCM?:** NCM is a method of initializing the weights of the classifier that is preferable to a random initialization. Compared to training a linear classifier, it can be done in a single round and is not affected by heterogeneity. Furthermore, in the FedNCM+FT approach we use FedNCM to initialize a standard softmax classifier.
>
> **Question 4:**
> Fig. 1 is discussed in L255-L272 which emphasizes when the behaviors differ across datasets. We will further briefly expand the discussion of the initial drop in accuracy on EuroSAT (see more detailed reply to reviewer tHeD).
>
> **Regarding Limitations:** We will expand the limitations to mention that we do not provide guarantees on more rapid convergence of the two phase process, note however that we do inherit the guarantees of the base FL method. Regarding large scale datasets, we emphasize above we use realistic and commonly used transfer learning datasets.

---

> > ### Comment · Reviewer_8wWr · 2023-08-16
> > **Thank you for the response**
> >
> > Thank you for your comprehensive response. I understand now the difference compared to Nguyen et al ICLR 2023. Overall I think fine-tuning is more important than NCM, from the experimental results. For example, on CIFAR-10 FedNCM + FT only improves FT by less than 2%. On other datasets like Flowers, however, the improvement of FedNCM on FT is more significant. I think it is interesting to understand when and how FedNCM helps, although I appreciate the extensive experiments done in this work.
> >
> > Additional question: why is FedNCM + FT not implemented for CIFAR-10 $\times$ 32?

---

> > > ### Author Response · Authors · 2023-08-17
> > > **Communication, Compute cost and Robustness to Heterogeneity are greatly improved by FedNCM+FT**
> > >
> > > Thank you for the follow-up. We want to emphasize that our FedNCM+FT does not just improve final accuracy (as noted quite substantially in some datasets) but also critically the convergence speed over naive FT which leads to large gains in communication and compute cost particularly under a budget. In addition we obtain larger performance gains in highly heterogeneous settings (see discussion with reviewer 1Br4). The NCM is critical for obtaining these large advantages since it is a negligible cost (and robust to heterogeneity) initial step. Communication, compute cost, heterogeneity are all essential performance factors in FL.
> > >
> > > Note that in Table 1 for fairness of comparison we run the FT and FedNCM+FT both for the same amount of rounds even if FedNCM+FT has converged, thus the total compute cost is not representative of the drastic advantages in communication that are seen when observing Figure 1. We refer the reviewer to consider the large gains observed in FT vs FedNCM+FT in Figure 1 and their practical consequences in FL. We will clarify this further in the text.
> > >
> > > Regarding CIFAR-10 x 32, the results for that shown in Tab 1 are taken from Nguyen et al (as clarified in L238-242) to directly show that when doing the native model resolution the FT acc is much higher and the gap of  FT and LP is smaller (see L238-241).  Thus in all our experiments we use CIFAR-10 upsampled to 224x224 (as done in all other datasets). We will more clearly indicate the CIFAR-10 x 32 results are added from Nguyen et al. as reference and add the FedNCM+FT result for this case which we have run now giving 67.9% (vs 63% for naive FT)

---

> > > > ### Comment · Reviewer_8wWr · 2023-08-18
> > > >
> > > > Sorry I don't understand the argument. As far as I understand the clients have to compute all the feature embeddings on the local training datasets, which can involve heavy computation if the training set is large. Also, I think the advantage of NCM in FL is due to the class imbalance in the datasets (in which case some other methods can be compared like Asymmetric loss e.g., https://arxiv.org/pdf/2106.03110.pdf). The authors should make this point more clearly.
> > > >
> > > > Given the new experiments I will update my score to 5. I still think this paper, though interesting, may need another round of revision.

---

> > > > > ### Author Response · Authors · 2023-08-18
> > > > > **FedNCM requires only one forward through all data vs many rounds and multiple local epochs of forward backward (FT)**
> > > > >
> > > > > Thanks for considering our rebuttal and increasing the score. We are happy to clarify the efficiency of FedNCM. We want to emphasize that FedNCM (without FT) is negligible in communication and compute cost as compared to FT (the prior work), while already obtaining good results (that can be further improved by a 2 phase procedure). The large gap is because FedNCM requires just one forward through all data and sends only one time the parameters, while FT does multiple local epochs and rounds of computation and communication.
> > > > >
> > > > > We illustrate this with an example below:
> > > > >
> > > > > Consider a standard FL FT based training (as done e.g. in Nguyen et al), with 5 local epochs of training and 500 rounds. Let’s also denote F as the number of FLOPs for a forward pass on all data and for simplicity consider 2*F the number of FLOPs for a backward pass on all data.
> > > > >
> > > > > **Compute (NCM vs FT)**: Each round consists of all clients forwarding and backwarding 5 times on all data. The total compute cost of FT is 3*(local epochs)*(rounds)* F = 7500*F  **Thus NCM will cost only F Flops while FT will cost 7500 times more**. Note that from Table 2 NCM alone can already sometimes exceed FT (e.g. flowers). To improve this further the excellent NCM starting point then leads to rapid convergence if followed by an FT phase (leading to savings in comm cost over FT due to the rapid convergence)
> > > > >
> > > > > **Communication (NCM vs FT)**: If we compare simply NCM to FT; NCM requires 1 round of sending weights to the clients, clients return only their centroid averages which is typically negligible cost. On the other hand FT involves 500 rounds of sending the entire model to the clients and the clients sending the entire model back to the server.
> > > > >
> > > > > The reviewer is right that with respect to heterogeneity advantage of NCM (only one of multiple advantages!) this is best seen when label distributions vary amongst clients, the most common heterogeneity considered in the FL literature (e.g. Nguyen et al, Legate et al, McMahan et al). We will further emphasize this in Sec 3.2.

---

> > > > > > ### Comment · Reviewer_8wWr · 2023-08-19
> > > > > >
> > > > > > Thank you for future clarification. I understand FT takes more computation, but in the paper NCM works well only on top of FT. Therefore we could only say that the overhead is small compared to FT.

---

> > > > > > > ### Author Response · Authors · 2023-08-19
> > > > > > > **FedNCM and FedNCM+FT have large gains over FT given a compute and communication budget**
> > > > > > >
> > > > > > > - FedNCM does already exceed or match FT for Flowers and in the new experiments on AG News (see attached pdf in top level message). In other cases it would be the best performer if the application has a constrained compute/comm budget
> > > > > > > - Our two phase (FedNCM +FT) leads to large gains in compute and communication without ever losing acc over FT. For example observing Figure 1 for CIFAR-10 FedNCM+FT achieves the final accuracy of FT in 60% less rounds, for Flowers in 99.5% less rounds, for CUB in 90% less rounds, for Cars in 75% less rounds, for Eurosat in  50% less rounds

---

### Official Review · Reviewer_63Tk · 2023-08-01

**Soundness:** 3 good
**Presentation:** 2 fair
**Contribution:** 2 fair
**Rating:** 4
**Confidence:** 3

**Summary:**

This paper studies the effect of tuning the linear head on federated feature learning. They show that well-tuning the linear head will make the base feature extractor converge closer to the optimal solution while reducing the total communication cost in FL. They evaluate their method on multiple FL datasets and show promising results.

**Strengths:**

Investigating the effect of a linear head on the final FL model tuning is interesting.


**Weaknesses:**

One thing is missing. When learning the linear head in the first stage, it is possible to get the closed-form solution of V. Why not do that? This will give a smaller upper bound for the right-hand side of Eq (2).

There are some works to improve the compute/communication efficient FL, such as https://arxiv.org/pdf/2206.08671.pdf (not limited to this). Some comparisons should be conducted.




**Questions:**

In Eq (2), the triangular inequality may be valid for a single client. Will this still be valid considering the FedAvg/FedAdam aggregation for w?

Can you put a theoretical convergence analysis of the two-stage FedNCM to show its advantage compared with the one without first-stage head tuning?

**Limitations:**

Please see above.

---

> ### Author Rebuttal · Authors · 2023-08-09
>
> We thank the reviewer for their helpful comments and their interest in our research direction. We address the comments in turn:
>
> **Weakness 1: Closed form solution**.
> Our experiments largely consider the cross entropy loss for which there is no closed form solution. Indeed, it is an interesting idea to obtain **V** by solving directly the right-hand side of the equation on the top of page 5. However, we emphasize that FedNCM gives a **V** which is the NCM classifier across all client datasets. While a closed form **V** for each client would lead to a question of how to aggregation a single **V** for all clients, since the **V**’s may heavily disagree.
>
> Finally, we emphasize that the analysis at the end of 3.3 is aimed to provide intuition for improved convergence and the loss we optimize is different.
>
> **Weakness 2: related work**
> We emphasize that our work is focused on compute/communication efficient FL *in the setting where pretrained models are available* which has different considerations. We thank the reviewer for the relevant reference, we will add it to our citations.  Shysheya et. al. focuses on fine-tuning methods based on adapting only the affine parameters with applications in few shot transfer learning, personalization, and also federated learning. Indeed the federated learning experiments focus on only CIFAR100 and the few shot federated learning setting. Our work, on the other hand, focuses exclusively on the federated learning setting and starts from the observations in very recent literature (ICLR 2023) regarding how to treat pre-trained models in federated learning. The fine-tuning method proposed in Shysheya et. al. can potentially be combined as part of the FT step with our two phase procedure to further reduce communication cost in the FT step; however, we leave this to future work. We also note that Shysheya is specialized to image data while our work is more general and can be applied to any classification problem, as shown in the new NLP experiments (see pdf)
>
> **Question Triangular inequality extension to aggregation and theoretical convergence results**
> The triangle inequality can be extended to multiple clients, with some additional assumptions. We consider below an extension to FedAvg style aggregation
>
> Consider the aggregation $\bar{w}=\frac{1}{K}\sum_k w_k$
>
> We can now write $|f(\bar{w}; [X_1, ... , X_K])-f(w^*; [X_1, ... , X_K])| \leq |f(w; X_k)-f(w^*; X_k)|$
>
> Then using an additional triangular inequality:
> $\leq \sum_k |f(w^*; X_k)-f(w_0;X_k)| + \sum_k |f(w_k; X_k)-f(w_0;X_k)| + \sum_k |f(\bar{w}; X_k)-f(w_k;X_k)|$
>
> With additional regularity assumption (namely L smoothness, and boundedness of both the data and the gradient) the last term can be bounded by a term proportional to $\sum_k |w_k-\bar{w}|$
>
> The term $\sum_k |w_k-\bar{w}|$ can be seen as a measure of consensus between workers and existing works provide potential upper bounds.  For instance, using Lemma 10 of Stitch et al. [1],  this can be upper bound by several additional assumptions, notably small learning rates. Note that In the NTK regime assumptions of [Ren et al 2023], step sizes are relatively small, which matches
> Lemma 10 of Stitch et al. [1]. Bounding some notion of agreement between workers *e.g.* $\sum_k |w_k-\bar{w}|$  is important to guarantee convergence (*e.g.* [1,2]).
>
> We emphasize the above is aimed to provide intuition and the analysis is limited to the first round of training, extending it beyond the first round and to obtain a rate is not obvious without additional theoretical tools. We emphasize there is a limited number of federated learning from pre-trained model references currently available and to our knowledge prior work in this area has focused on empirical analysis. Similarly,  two stage fine tuning procedures have been analyzed in terms of their generalization properties (Kumar et al, Ren et al) but not in terms of convergence rates. A deeper look at the theoretical convergence guarantees of federated transfer learning and two stage finetuning is certainly a valid and interesting research direction but is currently out of scope of our work which focuses on empirical results.
>
> [1] Stich, Sebastian U. "Local SGD converges fast and communicates little." arXiv preprint arXiv:1805.09767 (2018).
>
> [2] Reddi, Sashank, et al. "Adaptive federated optimization." arXiv preprint arXiv:2003.00295 (2020).

---

### Author Rebuttal · Authors · 2023-08-09

We thank the reviewers for their helpful comments and suggestions that help us improve the paper. We have responded to the reviewers in the individual comments.

We additionally include a pdf with NLP experiments suggested by reviewer tHeD. These show additional results on NLP Datasets using the Distillbert model as in (Nguyen et al). We consider two downstream datasets taken from the HuggingFace datasets repository;  AG News which consists of 4 classes, and a binary task Rotten Tomatoes. AG News has a total of 84K samples and  Rotten Tomatoes a total of 5K samples. We observe that FedNCM provides a powerful initial guess at low communication cost, that improves relative to FT as the dataset size decreases. We also observe the improved convergence performance of FedNCM+FT over FT as in our image experiments.

We also point the reviewers to our additional results with FedProx given in response to reviewer 8wWr

---

### Decision · Program_Chairs · 2023-09-21

**Decision:**

Accept (poster)

**Comment:**

The submission presents a simple recipe of practical relevance that has not appeared in the literature in this form before. The authors’ response clarified a number of reviewer questions regarding the described setting, the relation to prior work and the experimental evaluation. After the discussion phase, a majority of reviewers recommended acceptance.